# Inequalities of Infant Mortality in Ethiopia

**DOI:** 10.3390/ijerph20126068

**Published:** 2023-06-06

**Authors:** Nasser B. Ebrahim, Madhu S. Atteraya

**Affiliations:** 1Department of Public Health, Keimyung University, Daegu 42601, Republic of Korea; 2Department of Social Welfare, Keimyung University, Daegu 42601, Republic of Korea

**Keywords:** infant mortality, inequality, Ethiopia, Africa

## Abstract

(1) Background: Infant mortality is viewed as a core health indicator of overall community health. Although globally child survival has improved significantly over the years, Sub-Saharan Africa is still the region with the highest infant mortality in the world. In Ethiopia, infant mortality is still high, albeit substantial progress has been made in the last few decades. However, there is significant inequalities in infant mortalities in Ethiopia. Understanding the main sources of inequalities in infant mortalities would help identify disadvantaged groups, and develop equity-directed policies. Thus, the purpose of the study was to provide a diagnosis of inequalities of infant mortalities in Ethiopia from four dimensions of inequalities (sex, residence type, mother’s education, and household wealth). (2) Methods: Data disaggregated by infant mortalities and infant mortality inequality dimensions (sex, residence type, mother’s education, and household wealth) from the WHO Health Equity Monitor Database were used. Data were based on Ethiopia’s Demographic and Health Surveys (EDHS) of 2000 (*n* = 14,072), 2005 (*n* = 14,500), 2011 (*n* = 17,817), and 2016 (*n* = 16,650) households. We used the WHO Health Equity Assessment Toolkit (HEAT) software to find estimates of infant mortalities along with inequality measures. (3) Results: Inequalities related to sex, residence type, mother’s education, and household wealth still exist; however, differences in infant mortalities arising from residence type, mother’s education, and household wealth were narrowing with the exception of sex-related inequality where male infants were markedly at a disadvantage. (4) Conclusions: Although inequalities of infant mortalities related to social groups still exist, there is a substantial sex related infant mortality inequality with disproportional deaths of male infants. Efforts directed at reducing infant mortality in Ethiopia should focus on improving the survival of male infants.

## 1. Introduction

The pursuit of health equity and health justice places significant importance on child health and wellbeing [1,2,3]. In this context, it is essential to prioritize child health and reduce child health inequalities, including infant mortality, to achieve health equity [2], as higher rates of infant mortality challenge health justice and health equity goals. Furthermore, health justice and health equity have been influenced by social factors related to where individuals are born, live, work, and age [4,5,6,7]. Therefore, addressing social determinants of health is crucial to achieving health equity and reducing health inequalities, including inequalities in infant mortalities. Thus, the context-specific exploration of inequality in infant mortality is needed, which identifies determinants of health inequality, and further provides policy makers with insight into designing policies to enhance health justice so that every child has an equal opportunity to achieve good health [8,9,10,11]. In this context, we took the example of Ethiopia to examine inequalities in infant mortality using nationally representative samples.

The infant mortality rate is the number of deaths per 1000 live births in children under 1 year of age [12]. Inequality in infant mortality rates not only affect population health status, but also suggest how well a country can provide basic health services, access health care, nutritional status, access to the essential vaccinations, and overall social development [13,14]. Although infant mortality rates have been continuously decreasing globally, in 2021, 5 million of all under-5 deaths occurred within the first year of children’s lives [15]. Many of these children died in resource-poor countries because of deprivations, including lack of quality health care, incomplete vaccination, lack of clean water and sanitation, and from preventable infectious diseases. Furthermore, the inequality in child mortality rates reflects the wide-range of socio-economic differences that contribute to the global child health disparities. For example, Sub-Saharan African countries are attempting to reduce the current average under-5 mortality rate of 73 deaths per 1000 live births, which is more than twice the global rate (28 per 1000 live birth) and is nearly 24 times bigger than the under-5 mortality in the Republic of Korea (3 per 1000 live birth) [15].

The United Nations Sustainable Development Goal 3 (UN SDG 3) is a global development agenda aimed to strengthen “healthy life for all ages”. Notably, the United Nations Sustainable Development Goal 3.2 (UN SDG 3.2) sets targets to reduce preventable neonatal and under-5 deaths to 12 and 25 deaths per 1000 live births, respectively, by 2030 [16]. Besides, reducing inequalities within and among countries is one of the sustainable development goals (SDG 10). 

The infant mortality rate in Ethiopia in 2016 was 48 deaths per 1000 live births [17]—a significant drop from 97 deaths per 1000 live births in 2000 [18]. Moreover, Ethiopia has made substantial progress in achieving most of the maternal and child-related Millennium Development Goals (MDGs), and as such, there was a 67% reduction in under-5 mortalities and a 71% decline in the maternal mortality ratio [19]. This is partly the result of the Ethiopian government’s policy innovations and initiatives (i.e., the Health Extension Program, 2003, and the Health Sector Development Program IV, 2010–2015) to improve health equity in the country. Nevertheless, Ethiopia has significant inequality in infant mortalities [17]. 

The causes of inequities in health, including children’s health, have social roots and are often referred to as social determinants of health. “Social determinants of health are the conditions in the environments where people are born, live, learn, work, play, worship, and age that affect a wide range of health, functioning, and quality-of-life outcomes and risks [20]”. These conditions (i.e., sociocultural, economic, and environmental factors) may affect the physical and mental health and overall wellbeing of individuals or communities. These conditions can significantly impact child health equity and reduce child mortality.

Marmot and Wilkinson [20] suggested that social policy should address inequality factors and strive to eliminate health inequities for all, and promote health equity and justice. Since the distribution of infant deaths is not equal across segments of the Ethiopian population [10,21], further assessment of infant mortality inequalities is important to identify severely affected or disadvantaged social groups, which may also help inform equity-directed policies and programs and close inequality gaps in infant health [22], as well help achieve SDGs through interventions specifically targeted at socially disadvantaged groups [1,23,24]. 

The most recent literature on child mortality in Ethiopia has highlighted that high infant mortality is still prevalent—albeit there is a decrease in infant mortality compared to past years—and social and economic factors remain risk factors for high infant mortality rates [25,26,27,28,29,30,31]. For example, two studies [28,31] have examined factors that are associated with childhood illness and under-five mortalities. Tessema et al. [28] reported that under-5 mortality decreased between 1990 and 2019; however, were unevenly distributed among Ethiopian regions. Higher infant mortality rates were reported for Dire Dawa city and Somali region. Similarly, Chilot et al. [31] reported that common childhood illnesses (i.e., acute respiratory infections, diarrhea, and fever) were associated with under-five mortality, and varied by regions (i.e., higher in Tigray, Northern Amhara, and SNNPR regions). Variables related to risks for common childhood illness included children living in rural residences, low birth weights, high community poverty, mothers’ age 35–49, and household socio-economic status. 

As mentioned earlier, high infant and under-five mortalities remain significant threats to children’s wellbeing and survival. Although previous studies have highlighted the significance of children’s health, further studies are needed to illuminate the unfinished agenda of achieving child health equity. Thus, the current study aims to examine infant mortality inequalities arising from four dimensions of inequalities (sex, residence type, mother’s education, and household wealth) from nationally representative data in Ethiopia. 

## 2. Materials and Methods

Data disaggregated by infant mortalities and infant mortality inequality dimensions (sex, residence type, mother’s education, and household wealth) from the WHO Health Equity Monitor Database (HEAT, 2021) were used [22]. The data used in the study were based on Ethiopia’s Demographic and Health Surveys (EDHS) of 2000 (*n* = 14,072), 2005 (*n* = 14,500), 2011 (*n* = 17,817), and 2016 (*n* = 16,650) households [17,18,32,33]. We used the WHO Health Equity Assessment Toolkit (HEAT) software to find estimates of infant mortalities along with inequality measures [22]. This tool has been widely used in previous studies to identify health inequalities in context-specific situations to monitor and assess progress, design cost-effective interventions, make governments accountable for addressing inequalities, and ultimately aims to promote health equity [34,35,36,37,38,39]. 

We compared inequalities in infant mortalities in children (male vs. female; rural vs. urban residence, educational levels of mothers (no education, primary education, secondary, and higher education), and household wealth (poorest, quantile 1, through richest, quantile 5). The first two inequality dimensions (sex and residence type) have subgroups that are binary, meaning inequalities are compared between two groups with specific reference groups (female and urban), respectively, by using difference (D), which is an absolute inequality measurement that represents difference in health between two groups. D is calculated by estimating the difference between the highest and lowest values [22]. In our study, we had two binary dimensions of inequalities to compare (male vs. female) and (rural vs. urban). Since infant mortality is an adverse indicator [22] D was calculated from infant mortality rate difference (male mortality rate minus female mortality rate) and (rural mortality rate minus urban mortality rate) in the disaggregated data. If no inequality existed, then D would be zero. Larger absolute D means greater inequality. Positive D means concentration of infant mortality in the disadvantage group (in males or in rural residents) and negative values indicate concentration of infant mortality in the advantaged group (in females or in urban residents) [22]. 

When dimensions of inequalities are compared in more than two groups, they are either ordered or non-ordered [22]. Educational levels and household wealth were used to compare inequalities in subgroups that were ordered from most disadvantaged to most advantaged subgroups (e.g., ordered from no education to primary, secondary and higher education, and household wealth was ordered from poorest quantile 1 to the richest quantile 5). For these dimensions of inequalities (for simplicity and clarity) we used absolute concentration index (ACI), which measures health gradient among subgroups in a population, and to reflect the degree to which inequality in infant mortality is concentrated in different subgroups [22]. Zero ACI value means there is no inequality; positive numbers show a concentration of health indicator among the advantaged groups; and negative values indicate concentration of health indicator in the disadvantaged group. Higher ACI means larger inequalities [22].

## 3. Results

Overall, the infant mortality rate in Ethiopia has decreased from 124.4 and 100.6 deaths per 1000 live births in 2000 to 74.2 and 46.8 in 2016, for male and female infants, respectively, representing a decrease of 40.4% among females and 53.5% among males. The decrease was, however, incremental for both sexes over the years (Figure 1), with male infants consistently having higher infant mortality rates than the females (Figure 1 and Figure 2). In 2000, the absolute difference (D) in mortality rates among female and male infants was 23.8 deaths per 1000 live births in favor of females. However, in 2016, although both sexes had the lowest infant mortalities in 16 years, the inequality between female and male infant mortalities was the largest. Absolute sex related difference (D) in morality was 27.5 deaths per 1000 live births in favor of female infants (Figure 2).

When infant mortality rates among females and males were compared in 14 Sub-Saharan African countries, including 3 immediate neighbors of Ethiopia—Kenya, Sudan and South Sudan—Ethiopia had the largest inequalities in infant mortalities between the sexes, while Mozambique had the smallest inequality between the sexes. In all the settings, however, male infants had higher infant mortality rate than female infants (Table 1).

Regarding place of residence, infant mortality related to this dimension of inequality has decreased between 2000 and 2016—a reduction of 46% for rural residents and 44% for urban residents (Figure 3). However, infants residing in rural areas, had consistently higher infant mortality rates than infants residing in urban areas in Ethiopia (Figure 4). Place of residence related inequality in infant mortality decreased between 2000 and 2016 except in 2011. Inequality in infant mortality rate related to place of residence deceased from 18.2 deaths per 1000 live births in 2000 to 8 deaths per 1000 live births in 2016, the lowest difference (Figure 4). 

Similarly, regardless of the level of education of mothers, infant mortality has declined over the years for all infants (2000–2016). The highest decline (46.3%) was for infants whose mothers had no formal education. Nonetheless, infants whose mothers had no formal education had the highest infant mortality rate consistently, although the inequality had shrunk by 2016 (Figure 5). Mother’s education related to inequality as measured by absolute concentration index (ACI) showed that infant mortality was concentrated among infants whose mothers had no formal education (Figure 6). The inequality related to mother’s education declined from 5.4 deaths per 1000 live births in 2000 to 2.3 deaths per 1000 live births in 2016 (Figure 6). 

Household wealth quantile or economic status related inequality in the study year (2000–2016) showed that infants from wealthiest households (quantile 5 or richest 20%) had the lowest infant mortality except in the year 2000 (Figure 7). Specifically, in 2011, infant mortality was concentrated among poor households (quantile 1 or poorest 20%) as shown by the absolute concentration index (ACI). However, inequalities of infant mortality related to household wealth had virtually disappeared in 2016 (Figure 8).

## 4. Discussion

We analyzed four inequality dimensions of infant mortality in Ethiopia (sex, residence type, mother’s education, and household wealth). Our analysis has revealed that inequalities related to most of these factors still exist; however, differences arising from residence type, mother’s education, and household wealth were narrowing with the exception of sex-related inequality where male infants were markedly at a disadvantage. Consistent with the global trend of improving survival and declining infant mortalities [15], overall infant mortality in Ethiopia has significantly decreased over the years. However, we found significant inequalities between the sexes. The absolute difference between male and female infant mortalities from 2000 to 2016 ranged between 21.1 and 27.5 deaths in favor of female infants. Regardless of extensive documentation of gender-based inequalities (i.e., sex of the child) in infant mortality in a global context in the previous literature [40,41,42], the significance of differences in infant mortality rate, trend, and patterns is wildly varied and related to intricate social and economic factors within and among countries [40], including the use of new medical therapies.

For example, in the United States of America (U.S.A.), in 2019, the female infant mortality rate was 5.1 deaths per 1000 live births, while for male infants it was 6 deaths per 1000 live births. In the U.S.A., in 1970, the gap in infant mortality between the sexes was even wider (22.3 and 17.3) deaths per 1000 live births for male and female infants, respectively [43]. The remarkable decline in infant mortalities and the narrowing of the gap between the sexes in the U.S.A. was attributed to significant advances in neonatal care and availability of new therapies [43]. Comparing difference between male and female infant mortalities in Ethiopia with 14 Sub-Saharan African children population, including Ethiopia’s neighboring countries, the difference between the sexes was largest in Ethiopia. The disadvantage of male infants may be related to perinatal conditions that disproportionately affect male infants [40]. For example, male infants have been reported to have weak immune response and higher deaths from infections than female infants [44]. Additionally, as compared to females, males are more likely to be preterm at birth and develop respiratory distress syndrome (RDS) [41,45]. Males also tend to suffer more birth complications that increase the risk of adverse outcomes [46]. These natural survival disadvantages of male infants are, however, inadequate to explain the excess male infant mortality in Ethiopia. For example, despite comparable socioeconomic conditions, Ethiopia has 12-times, 4.7-times, and 4-times higher infant mortality differences between the sexes than Mozambique, Rwanda, and Kenya, respectively (Table 1). Nonetheless, the wide gap in infant mortality between the sexes could in part be explained by differences in neonatal mortality in Ethiopia. According to the report from 2016 Demographic and Health Survey, the neonatal mortality was 49 deaths per 1000 live births for male infants and 26 for female infants (difference: 23 deaths per 1000 live births). For the postnatal period, it was 26 deaths for males vs. 20 deaths for females per 1000 live births, representing 50% less deaths for male infants during this period [17]. Thus, it is important to reduce excess male infant mortality in Ethiopia through improved perinatal care, access to obstetrics care (not just skilled birth attendants) including expansion of neonatal intensive care units and availability of therapy for preterm babies. 

Although inequalities in infant mortality related to residence type, mother’s education, still exist in Ethiopia, the gap between disadvantaged and advantaged groups was narrowing. However, in some of the inequality dimensions examined, substantial inequalities and decreased infant mortality rates were observed. Mckinnon et al. [23], in their multicounty study of neonatal mortality, reported similar phenomenon and attributed the increased inequalities and decreased neonatal mortalities to the rich and well-educated women residing in urban areas initially benefitting from improved perinatal, obstetric care, and delivery services. Similarly, we observed infant mortality inequalities related to residence type, mother’s education, and household wealth later declined (e.g., in 2016) as disadvantaged groups may have access to improved health services as well [23].

Our results concur with previous studies [10,47,48,49,50,51,52] that child health inequality exists, and it is crucial to address it along with associated risk factors. These studies [10,47,48,49,50,51,52] also provide future directions for enhancing child health equity in Africa, particularly in Ethiopia that include advancing support to communities in rural and regional states, improving gender equality, and developing a partnership with diverse stakeholders. In the Ethiopian context, this means that enhancing health equity and reducing infant mortality requires commitment of resources. However, such efforts could be undercut by lack of resources, especially in areas affected by recurrent droughts and food shortages that severely affect children and the poor. In the short term, community capacity building at grassroot level in tandem with raising awareness about infant mortality inequities is essential. In the long term, economic development and health policy alignment of the needs of population could play a role in reducing infant mortality in Ethiopia. 

The study has some limitations, as variables such as ethnicity, child raising traditions, and access and availability of health service were not included in the study. Nevertheless, we found that inequality between female and male infant mortalities was the largest, with male infants being at disadvantage. Likewise, higher infant mortality rates were observed for children in rural areas. Infants whose mothers had no formal education consistently had the highest infant mortality rate, although the inequality gradually shrunk over the years. These findings further provide an insight into “health justice issues” and the need to enhance health equity efforts, especially among male infants, residing in rural areas whose mothers had no formal education. 

## 5. Conclusions

In Ethiopia, inequalities of infant mortalities related to social groups exist. However, differences arising from socioeconomic factors such as residence type, mother’s education and household wealth were narrowing. But, there is a substantial sex-related infant mortality inequality with disproportional deaths of male infants that is entirely unexplainable by the natural survival advantage of female infants. Thus, it is important to raise awareness about the unnecessary loss of lives among male infants in Ethiopia. Improved child-rearing methods that focus on child survival such as breastfeeding, immunization, prevention of infections, and malnutrition should be strengthened. Extra efforts should be directed at improving survival of male infants which may help Ethiopia achieve Sustainable Development Goals (SDG) 3.2., although, at the moment, it is a challenge for the country because of the drought and food shortages. Nonetheless, it is important to provide optimum child-rearing methods focused on proper nutrition, safety, prevention of infections, and early clinical interventions to enhance children’s health. Equitable economic development coupled with health policy targeting the most disadvantaged group may mitigate inequalities in infant mortalities in Ethiopia in the long run. 

## Figures and Tables

**Figure 1 ijerph-20-06068-f001:**
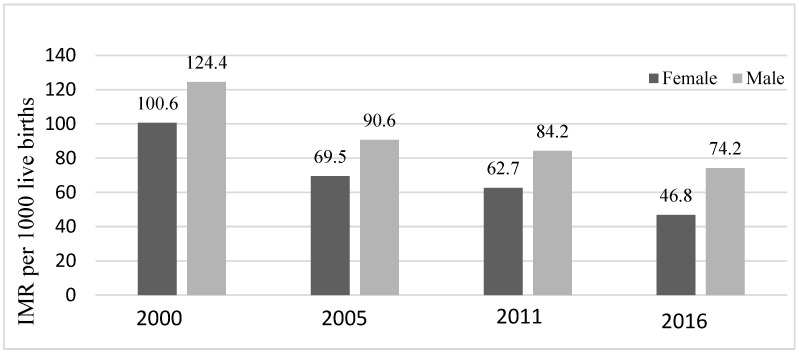
Infant mortality rates per 1000 live births among Ethiopian children disaggregated by infants’ sex (DHS* 2000, 2005, 2011, 2016) Note: DHS* = Demographic and Health Surveys; IMR = Infant Mortality Rates.

**Figure 2 ijerph-20-06068-f002:**
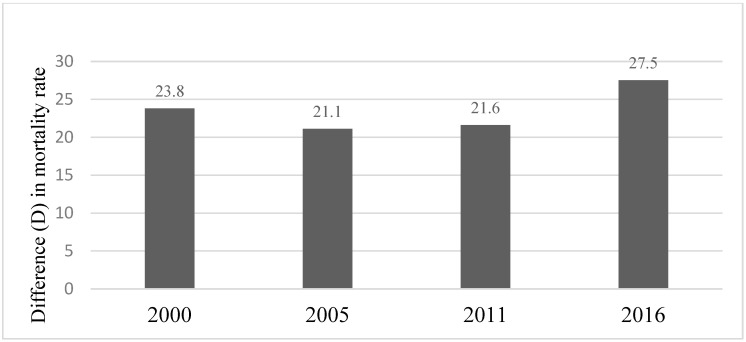
Inequality in infant mortality rate related to the infant sex: Difference (D). (DHS* 2000, 2005, 2011, 2016) Note: * = Demographic and Health Surveys; Difference (D) = Male infant mortality − Female infant mortality.

**Figure 3 ijerph-20-06068-f003:**
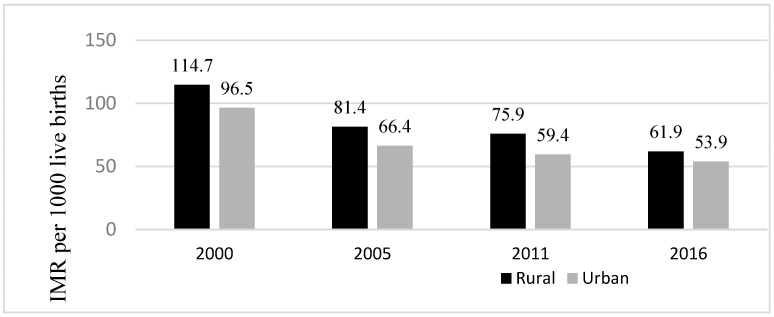
Infant mortality rates per 1000 live births among Ethiopian children disaggregated by infants’ place of residence (DHS* 2000, 2005, 2011, 2016) Note: * = Demographic and Health Surveys; IMR = Infant Mortality Rates.

**Figure 4 ijerph-20-06068-f004:**
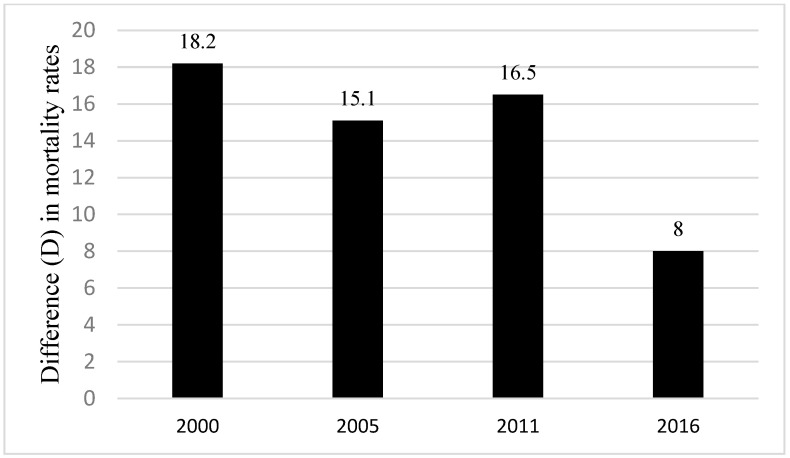
Inequality in infant mortality rates per 1000 live births related to the place of residence (DHS* 2000, 2005, 2011, 2016) Note: Difference (D) = Rural mortality − Urban infant mortality; * = Demographic and Health Surveys.

**Figure 5 ijerph-20-06068-f005:**
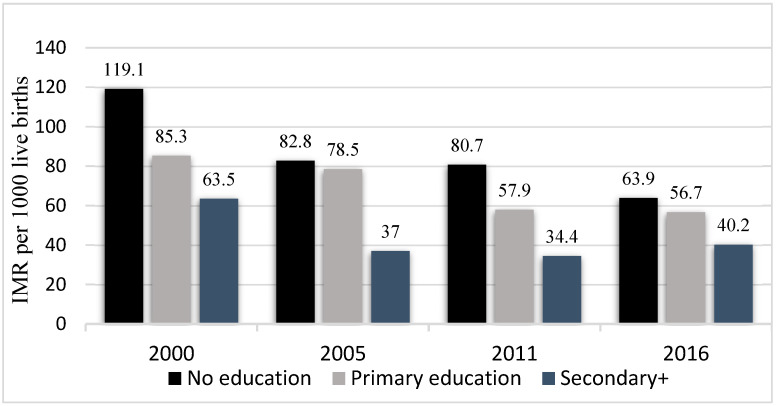
Infant mortality rate per 1000 live births disaggregated by mothers’ education (DHS* 2000, 2005, 2011, 2016) Note: IMR = Infant Mortality Rates; DHS* = Demographic and Health Surveys.

**Figure 6 ijerph-20-06068-f006:**
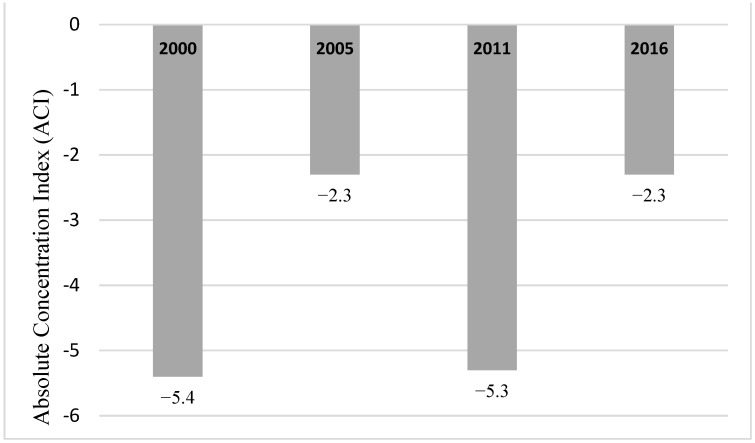
Mothers’ education related inequality in infant mortality: Absolute Concentration Index (ACI) (DHS*, 2000, 2005, 2011, 2016) Note: DHS* = Demographic and Health Surveys.

**Figure 7 ijerph-20-06068-f007:**
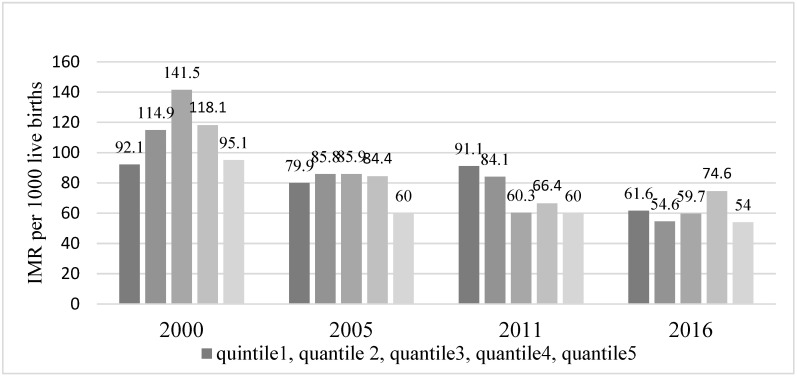
Infant mortality rate per 1000 live births desegregated by economic status (wealth quantile) (DHS* 2000, 2005, 2011, 2016) Note: IMR = Infant Mortality Rates; DHS* = Demographic and Health Surveys.

**Figure 8 ijerph-20-06068-f008:**
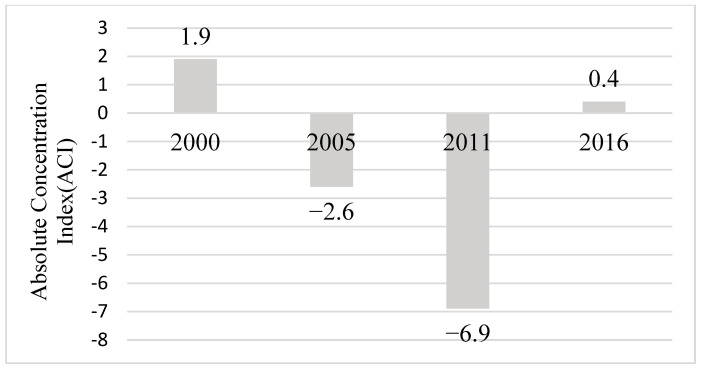
Household economic status (wealth quintile) related inequality in infant mortality: Absolute Concentration Index (ACI). (DHS* 2000, 2005, 2011, 2016) Note: DHS* = Demographic and Health Surveys.

**Table 1 ijerph-20-06068-t001:** Summary of absolute difference (D) in infant mortalities among males and females in 14 African countries.

Setting, Country	Survey Year	Indicator Name	Dimension	Summary Measure	Inequality Estimate
Ethiopia	2016	Infant mortality	Sex	Difference (D)	27.5
Burundi	2016	≡^im^	≡^s^	≡^d^	6.7
Chad	2014	≡^im^	≡^s^	≡^d^	15.1
Guinea	2016	≡^im^	≡^s^	≡^d^	7.9
Guinea-Bissau	2014	≡^im^	≡^s^	≡^d^	15.9
Malawi	2015	≡^im^	≡^s^	≡^d^	16.0
Mali	2015	≡^im^	≡^s^	≡^d^	17.8
Mozambique	2015	≡^im^	≡^s^	≡^d^	2.3
Rwanda	2014	≡^im^	≡^s^	≡^d^	5.9
Sierra Leone	2017	≡^im^	≡^s^	≡^d^	17.0
Uganda	2016	≡^im^	≡^s^	≡^d^	9.8
Sudan	2014	≡^im^	≡^s^	≡^d^	11.9
South Sudan	2010	≡^im^	≡^s^	≡^d^	13.6
Kenya	2014	≡^im^	≡^s^	≡^d^	6.8

Note: Difference (D) = Male infant mortality — Female infant mortality; ≡^im^ = Indicator Name refers to Infant Mortality; ≡^s^ = Dimension refers to Sex; ≡^d^ = Summary Measure refers to Difference (D).

## Data Availability

Data are available from WHO Health Equity Monitor Database.

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
