# Peer review of "Inequalities of Infant Mortality in Ethiopia"

_ijerph, 2023, doi:10.3390/ijerph20126068_

Round 1
Reviewer 1 Report
CONSIDERATIONS:
The introduction is the part of the scientific article in which the author informs what was researched and why the investigation was carried out. I suggest that authors try to meet this demand of Scientific Writing.
It is a place to specify particular aspects of the research, such as the justification for its realization, the originality and the logic that guided the investigation. Some questions help in writing. What is the study about? Why was it done? Why should it be published? I've missed those answers throughout this Introduction Section.
It also seeks to show that the research is based on solid foundations. Thus, in the introduction, a link is made with the relevant literature. What was known about the subject at the beginning of the investigation? What was not known about the matter and motivated the investigation? Answering these questions involves a process of choosing the works to cite.
As it is an original article, there is no room for extensive review of what has been published on the subject. This is not a review article. However, if the author has carried out a detailed review of the literature, he should try to publish it separately and, in this case, present it in his final dissertation, especially with the table of records made.
An outcome is suggested at the end of the Introduction Section, before the Objective Section: among the criteria used to choose them are relevance, accessibility and timeliness.
Still, it is a section that must be covered from the general to the specific and with insertion of the Objective Section in its last paragraph and with adjustments to it, listed in the text.
The purpose of the publication is usually found at the end of the introduction. If the string of subjects at the beginning of the article is adequate, the objective will be the natural consequence and closing of the introduction. When starting the writing, it is convenient to have the objective of the article in writing. It will be the fulcrum for the composition of the entire text. Those who assess the quality of an article usually check if the text reflects the objective and, in particular, if the objective and the conclusion match. Hence the importance of keeping the objective in mind while writing.
There are several ways to express the objective. It can be related to the field of research, whether frequency, diagnosis, etiology, treatment and prevention of diseases. Another way is to write the objective according to the method used. In this case, the traditional uses of Epidemiology can be of support. However, the authors promise to EVALUATE (they use this verb) and in this case there is a need to perform a statistical analysis with the insertion of multiple regression to explain the results.
The first three parts of the body of an original article – introduction, method and results – are essential for the composition of this Discussion Section.
In this text, the discussion section is essential for the closing of the manuscript, as it completes the structure of the article.
Reviewer 2 Report
I find the script interesting but i find the introduction very lengthy which can be reduced.
line 45-46 "one of year of age" can be replaced by "one year of age"
line 285 "(difference: 13 deaths per 1000 live births)" can be replace by "(difference: 23 deaths per 1000 live births)"
Author Response
Reviewer 2
Comments and Suggestions for Authors
I find the script interesting but i find the introduction very lengthy which can be reduced.
- Thank you for your advice. We have added literature and discussions to meet the minimum word count required in the journal. Agreeing with you, we have tried to reduce the length throughout the manuscript.
line 45-46 "one of year of age" can be replaced by "one year of age"
- Thank you for your advice. We corrected it.
line 285 "(difference: 13 deaths per 1000 live births)" can be replace by "(difference: 23 deaths per 1000 live births)"
- Thank you for your advice. We corrected it.
Reviewer 3 Report
The paper presented is very interesting and is a good contribution to the scientific community. In fact, I found it particularly nice for its clarity of exposition. Without any doubt, it is a work that deserves to be published. The only drawback I could put to the work is related to the format. At some points there is a lot of separation between words and the fonts vary between the different figures. These aspects should be adjusted. Also, I would suggest that the images be a little larger so that they can be seen correctly. Otherwise, I repeat my satisfaction and congratulations for the work.
Author Response
Reviewer 3
The paper presented is very interesting and is a good contribution to the scientific community. In fact, I found it particularly nice for its clarity of exposition. Without any doubt, it is a work that deserves to be published. The only drawback I could put to the work is related to the format. At some points there is a lot of separation between words and the fonts vary between the different figures. These aspects should be adjusted. Also, I would suggest that the images be a little larger so that they can be seen correctly. Otherwise, I repeat my satisfaction and congratulations for the work.
- Thank you for your advice. We have corrected it throughout the manuscript.
Reviewer 4 Report
Dear researchers,
You have made a very good analysis, congratulations.
Although the indicators taken for the analysis of inequality in infant mortality are correct, they may not be sufficient to explain the inequality. I am of the opinion that indicators such as ethnicity, traditions, scope of service can play a role in inequality. Therefore, this study has limitations.
I am of the opinion that the situation should be given more clearly to the country's health policies and health services in the debate on reducing inequalities.
Author Response
You have made a very good analysis, congratulations.
Although the indicators taken for the analysis of inequality in infant mortality are correct, they may not be sufficient to explain the inequality. I am of the opinion that indicators such as ethnicity, traditions, scope of service can play a role in inequality. Therefore, this study has limitations.
I am of the opinion that the situation should be given more clearly to the country's health policies and health services in the debate on reducing inequalities.
- Thank you for your advice. We agree with you and added the following line in the manuscript (page number 8, line number 320-322)